# Hippocampal-occipital connectivity reflects autobiographical memory deficits in aphantasia

**Merlin Monzel[1,2]\*[†], Pitshaporn Leelaarporn[2,3]\*[†], Teresa Lutz[2], Johannes Schultz[4,5], Sascha Brunheim[2], Martin Reuter[1], Cornelia McCormick[2,3]**

[1]Department of Psychology, University of Bonn, Bonn, Germany; [2]German Center for Neurodegenerative Diseases, Bonn, Germany; [3]Department of Old Age Psychiatry and Cognitive Disorders, University Hospital Bonn, Bonn, Germany; [4]Center for Economics and Neuroscience, University of Bonn, Bonn, Germany; [5]Institute of Experimental Epileptology and Cognition Research, Medical Faculty, University of Bonn, Bonn, Germany

**\*For correspondence:**
monzel@uni-bonn.de (MM);
pitshaporn.leelaarporn@dzne.de (PL)

[†]These authors contributed equally to this work

**Competing interest:** The authors declare that no competing interests exist.

## Abstract

Aphantasia refers to reduced or absent visual imagery. While most of us can readily recall decade-old personal experiences (autobiographical memories, AM) with vivid mental images, there is a dearth of information about whether the loss of visual imagery in aphantasics affects their AM retrieval. The hippocampus is thought to be a crucial hub in a brain-wide network underlying AM. One important question is whether this network, especially the connectivity of the hippocampus, is altered in aphantasia. In the current study, we tested 14 congenital aphantasics and 16 demographically matched controls in an AM fMRI task to investigate how key brain regions (i.e. hippocampus and visual-perceptual cortices) interact with each other during AM re-experiencing. All participants were interviewed regarding their autobiographical memory to examine their episodic and semantic recall of specific events. Aphantasics reported more difficulties in recalling AM, were less confident about their memories, and described less internal and emotional details than controls. Neurally, aphantasics displayed decreased hippocampal and increased visual-perceptual cortex activation during AM retrieval compared to controls. In addition, controls showed strong negative functional connectivity between the hippocampus and the visual cortex during AM and resting-state functional connectivity between these two brain structures predicted better visualization skills. Our results indicate that visual mental imagery plays an important role in detail-rich vivid AM, and that this type of cognitive function is supported by the functional connection between the hippocampus and the visual-perceptual cortex.

## eLife assessment

This **important** work substantially advances our understanding of episodic memory in individuals with aphantasia, and sheds light on the neural underpinnings of episodic memory and mental imagery. The evidence supporting the conclusions is **convincing**, including evidence from a well-established interview paradigm complemented with fMRI to assess neural activation during memory recall. The work will be of broad interest to memory researchers and mental imagery researchers alike.

## Introduction

Our unique and personal memories are stored in autobiographical memories (AM) providing stability and continuity of our self (*Svoboda et al., 2006*). For most of us, travelling mentally back in time and re-visiting such unique personal events is associated with vivid, detail-rich mental imagery (*D'Argembeau and Van der Linden, 2006*; *Greenberg and Knowlton, 2014*). This vivid mental imagery during the re-experiencing of AMs has become a hallmark of autonoetic, episodic AM retrieval. However, up to date, it remains unclear to what extent episodic AM retrieval depends on visual mental imagery and what neural consequences a lack of mental imagery has on episodic AM retrieval. This knowledge gap exists because separating AM retrieval from mental imagery is a complex and challenging task.

One way to address this conundrum is to study people with aphantasia (*Zeman et al., 2015*). Recent research defines aphantasia as a neuropsychological difference in which people experience a marked reduction or complete lack of voluntary sensory imagery (*Monzel et al., 2022a*). This state is associated with psychophysiological alterations, such as reduced imagery-induced pupil contraction (*Kay et al., 2022*) and reduced imagery-induced priming effects (*Keogh and Pearson, 2018*; *Monzel et al., 2021*). Thus, aphantasics offer the unique opportunity to examine the consequences for episodic AM retrieval in the absence or marked reduction of voluntary imagery. Indeed, a handful of previous studies report convergent evidence that aphantasics report less sensory AM details than controls (*Dawes et al., 2020*; *Dawes et al., 2022*; *Milton et al., 2021*; *Zeman et al., 2020*), which may also be less emotional (*Monzel et al., 2023*; *Wicken et al., 2021*). Spatial accuracy, on the other hand, was not found to be impaired (*Bainbridge et al., 2021*). Yet, task-based functional activity has not been fully explored.

Neurally, the hippocampus has been established as a central brain structure to support the detail-rich episodic AM retrieval in the healthy brain (*Bauer et al., 2017*; *Brown et al., 2018*; *Burianova et al., 2010*; *McCormick et al., 2020*; *Moscovitch et al., 2005*), albeit some studies differentiate between hippocampal support for remote and recent autobiographical memories (see *Bayley et al., 2006*). In fact, hippocampal activity correlates with the vividness of AM recollection (*Addis et al., 2004*; *Sheldon and Levine, 2013*) and patients with hippocampal damage show marked deficits in detailed episodic AM retrieval (*Miller et al., 2020*; *Rosenbaum et al., 2008*). In addition, neuroimaging studies illuminate that the hippocampus is almost always co-activated with a wider set of brain regions, including the ventromedial prefrontal cortex (vmPFC), lateral and medial parietal cortices, as well as visual-perceptual cortices (*Svoboda et al., 2006*; *Addis et al., 2007*). Interestingly, especially during the elaboration phase of AM retrieval, when people engage in the active retrieval of episodic details to a specific AM, the hippocampus exhibits a strong functional connection to the visual-perceptual cortices, suggesting a crucial role of this connection for the embedding of visual-perceptual details into AMs (*McCormick et al., 2015*; *Leelaarporn et al., 2024*).

Yet, not many studies have examined the neural correlates of aphantasia, and none during AM retrieval. Of the little evidence there is, reports converge on a potential hyperactivity of the visual-perceptual cortices in aphantasia (*Fulford et al., 2018*; *Keogh et al., 2020*). A prominent theory posits that because of this hyperactivity, small signals elicited during the construction of mental imagery may not be detected (*Pearson, 2019*; *Keogh et al., 2020*). Pearson further speculates that since spatial abilities seem to be spared, the hippocampus may not be the underlying cause of aphantasia. In agreement, *Bergmann and Ortiz-Tudela, 2023* speculate that individuals with aphantasia might lack the ability to reinstate visually precise episodic elements from memory due to altered feedback from the visual cortex. In the same vein, *Blomkvist, 2023* proposes the extended constructive episodic simulation hypotheses (CESH+) that suggests that imagination and memory rely on similar neural structures, since both represent simulated recombinations of previous impressions. This hypothesis has been supported by shared representations for memory and mental imagery in early visual cortex (*Albers et al., 2013*; see also *Zeidman and Maguire, 2016*). Within this framework, the hippocampus is supposed to initiate sensory retrieval processes (e.g. in the visual-perceptual cortices; *Danker and Anderson, 2010*), comparable to its role in the hippocampal memory indexing theory (*Langille and Gallistel, 2020*). *Blomkvist, 2023* speculates that in aphantasics, either the hippocampal memory index or the retrieval processes may be impaired.

Thus, the main goal of our study was to examine the neural correlates of AM deficits associated with aphantasia. We hypothesized that the deficits in AM seen in aphantasia rely on altered involvement of the hippocampus, visual-perceptual cortices and their functional connectivity.

# Results

## VVIQ and binocular rivalry task

Aphantasics (*M*=16.57, SD = 1.02) scored significantly lower on the Vividness of Visual Imagery Questionnaire (VVIQ) than controls (*M*=62.94, SD = 8.71), $t(15.47)=21.12$, p<0.001, *d*=7.23. Furthermore, aphantasics and controls differed in the priming score of the binocular rivalry task, $t(18.04)=2.41$, p=0.027, *d*=0.87. While controls were primed by their own mental imagery in 61.3% (SD = 13.1 %) of the trials, aphantasics were only primed 52.6% (SD = 4.9 %) of the time. In fact, the performance of controls differed significantly from chance, $t(14) = 3.34$, p=0.005, *d*=0.86, whereas performance of aphantasics did not, $t(13) = 1.96$, p=0.072. Moreover, the VVIQ scores correlated positively with the performance on the binocular rivalry task, $r(28) = 0.43$, p=0.022. For the mock trials, no significant differences in priming scores were found between groups, $t(28) = 0.86$, p=0.396, or related to chance (aphantasics: $t(13) = 0.74$, p=0.475; controls: $t(15) = 0.42$, p=0.682). These findings validate our groups by indicating that visual imagery strength was diminished in aphantasics.

## Autobiographical interview

Regarding the Autobiographical Interview (AI), we found significant main effects of memory period, $F(1, 27)=11.88$, p=0.002, $\eta_p^2 = 0.31$, type of memory details, $F(1, 27)=189.03$, p<0.001, $\eta_p^2 = 0.88$, and group, $F(1, 27)=9.98$, p=0.004, $\eta_p^2 = 0.27$. When the other conditions were collapsed, aphantasics (M=26.29, SD = 9.58) described less memory details than controls (M=38.36, SD = 10.99). For aphantasics and controls combined, more details were reported for recent (M=35.17, SD = 14.19) than remote memories (M=29.06, SD = 11.12), and internal details (M=43.59, SD = 17.91) were reported more often than external details (M=20.64, SD = 8.94). More importantly, a two-way interaction was found between type of memory details and group, $F(1, 27)=54.09$, p<0.001, $\eta_p^2 = .67$, indicating that aphantasics reported significantly less internal memory details, $t(27) = 5.07$, p<0.001, *d*=1.83, but not significantly less external memory details, $t(27) = 0.13$, p=0.898, compared to controls (see *Figure 1b*). No two-way interaction between memory period and group, $F(1, 27)=0.62$, p=0.439, and no three-way interaction between memory period, group and type of memory details was found, $F(1, 27)=3.87$, p=0.060.

Based on the interaction effect between group and type of memory details, we compared specific categories of internal and external memory details between the groups. For internal details and in comparison to controls, aphantasics reported less internal events, $t(27) = 3.22$, p=0.016, *d*=1.17, less emotional details, $t(27) = 4.40$, p<0.001, *d*=1.59, less perceptual details, $t(27) = 4.95$, p<0.001, *d*=1.79, and less details regarding time, $t(27) = 5.27$, p<0.001, *d*=1.90, and place, $t(27) = 3.31$, p<0.013, *d*=1.20 (see *Figure 1c*). On the other hand, no significant differences were found for external details, including external events, $t(27) = 0.71$, p>0.999, semantic details, $t(27) = 0.02$, p>0.999, repetition, $t(27) = 0.46$, p>0.999, and other details, $t(27) = 0.45$, p>0.999 (see *Figure 1C*). Regarding the rating scales, we found that aphantasics showed less episodic richness, $t(27) = 7.50$, p<0.001, *d*=2.71, and less memory confidence, $t(27) = 5.85$, p<0.001, *d*=2.11 (see *Figure 1a*) as well as lower self-reported visualization scores, $t(27) = 11.92$, p<0.001, *d*=4.30, than controls.

## Debriefing questions

The debriefing questions were employed as a way for participants to reflect on their own cognitive abilities. Of note, these were not meant to represent or replace necessary future experiments. There were stark differences between the groups in how they answered our debriefing questions. While aphantasics reported that they typically have greater difficulty to recall autobiographical memories, $t(27) = 6.20$, p<0.001, *d*=2.31, and to use their imagination in daily life, $t(24) = 10.18$, p<0.001, *d*=3.93, they did not report difficulties in spatial orientation, $t(27) = 0.62$, p=0.541, *d*=0.23.

## Behavioral results of the fMRI AM task

We found stark differences for the vividness response between groups, $t(28) = 5.29$, p<0.001. While controls reported in 86% (SD = 26 %) of trials that their AM retrieval had been vivid, aphantasics indicated only in 20% (SD = 20 %) of trials that their AM retrieval had been vivid. Moreover, aphantasics responded slower (*M*=1.34 s, SD = 0.38 s) than controls (*M*=1.00 s, SD = 0.29 s) when they were asked whether their retrieved memories were vivid or faint, $t(28) = 2.78$, p=0.009, possibly reflecting uncertainty in their response. In contrast, there were no significant differences between groups during the

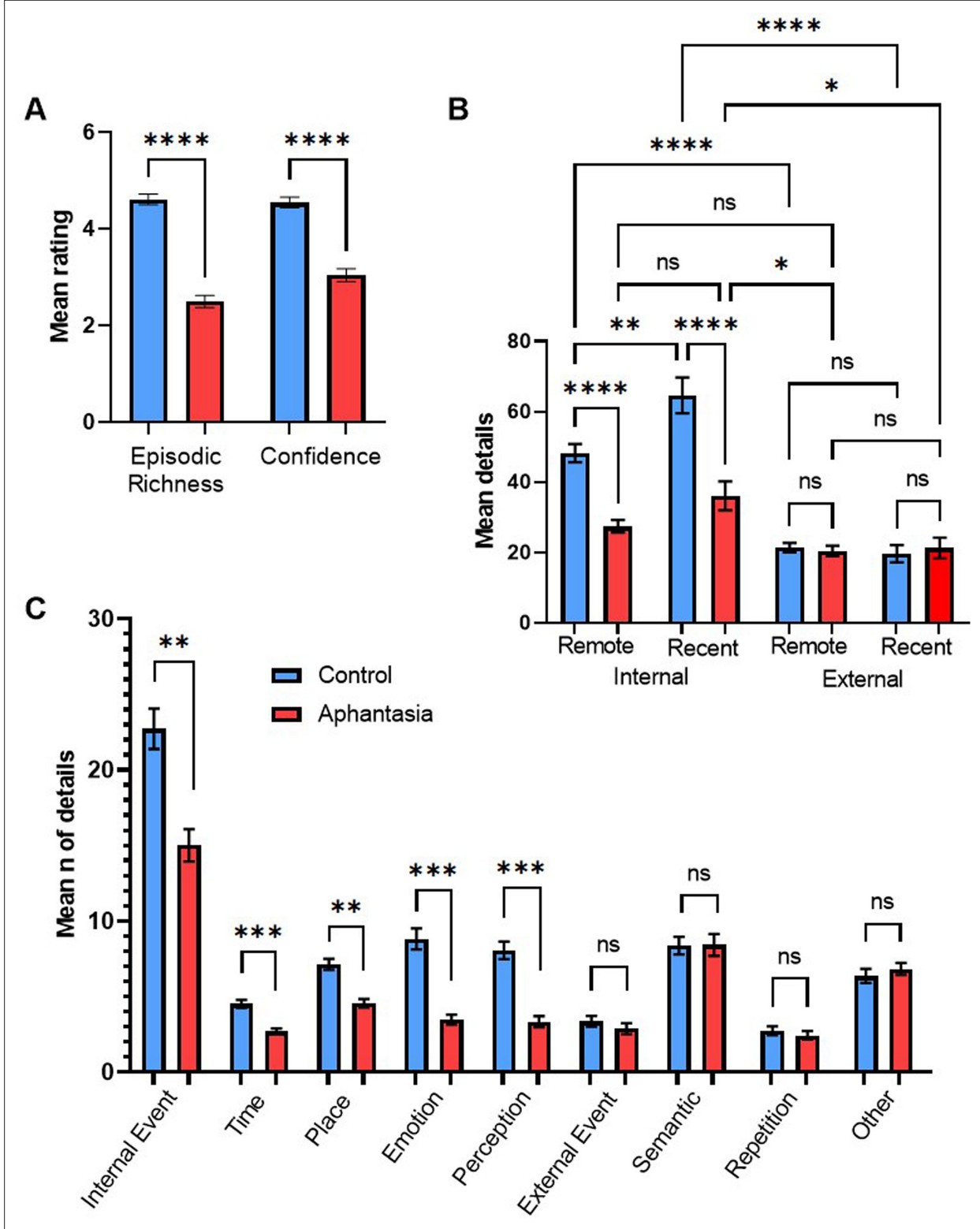

**Figure 1.** AM deficits associated with aphantasia. (**A**) Mean amount (± SEM) of episodic richness and confidence in the Autobiographical Interview for controls and aphantasics. (**B**) Mean amount (± SEM) of internal details and external details for recent and remote memories. (**C**) Mean amount (± SEM) of specific internal and external memory details for aphantasics and controls. * $p<0.05$, ** $p<0.01$, *** $p<0.001$, **** $p<0.0001$, n.s.=non-significant.

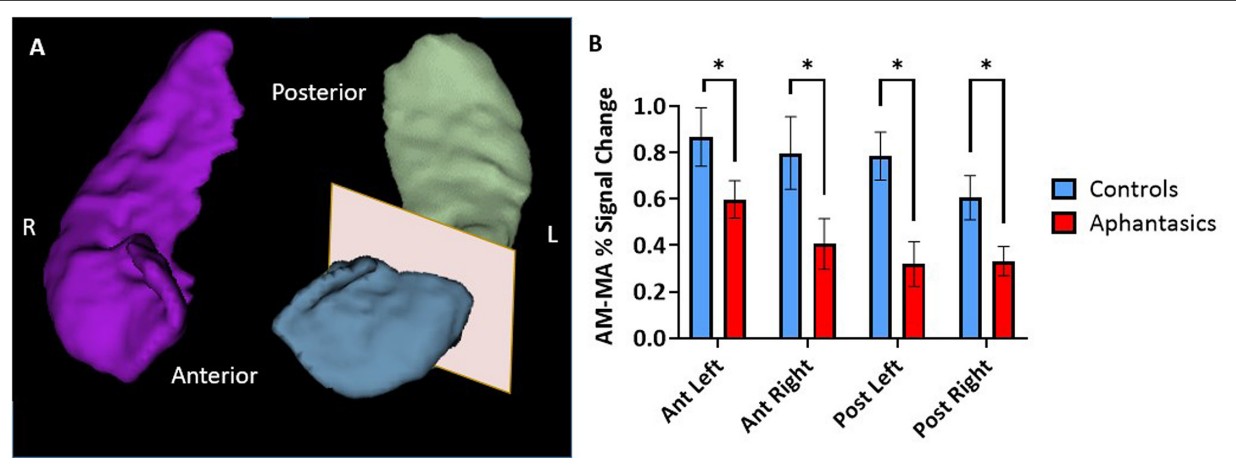

**Figure 2.** Reduced hippocampal activity during autobiographical memory retrieval associated with aphantasia. The signal intensities during autobiographical memory (AM) and mental arithmetic (MA) were extracted from anatomical hippocampal masks created from each individual participant. (**A**) An example of a 3D reconstruction of the hippocampus, separated into anterior and posterior portions for the left hippocampus. (**B**) The comparison between the percentage of signal change during the AM and MA tasks in the hippocampus of aphantasics and controls. Aphantasics show reduced differentiation between AM and MA than controls in all portions of the hippocampus. * p<0.05.

MA trials, neither on the easy/hard response, $t(28)$ = 1.16, p=0.255, nor on the reaction times, $t(28)$ = 0.58, p=0.567.

In addition, aphantasics and controls did not differ significantly in their time searching for a memory in AM trials, $t(19)$ = 1.03, p=0.315. On average, aphantasics spent 3.42 s (SD = 0.74 s) and controls spent 3.15 s (SD = 0.48 s). Furthermore, both groups did not differ in their speed to solve the math problems, $t(23)$ = 0.09, p=0.926. Aphantasics spent 3.87 s (SD = 0.97 s) to solve a math problem and controls spent 3.90 s (SD = 0.72 s). For the button press, there were 9% missing values in AM trials and 7% missing values in MA trials with no significant differences of missing values between groups, neither for AM trials, $t(19.98)$=1.11, p=0.281, nor for MA trials, $t(18.13)$=0.52, p=0.609.

### Native space differences in hippocampal activation during AM retrieval

First, we sought to examine the hippocampal activation during an established AM-fMRI-task in aphantasics and controls (see *Figure 2*). During fMRI scanning, participants saw either word cues (e.g. 'a party') and were asked to retrieve vivid, detail-rich AMs, or a number cue (e.g. 31+82) and were asked to solve the math problem. Using individual anatomical masks of the hippocampus, the extracted fMRI signals illustrated stark group differences in AM-associated activation, $F(17, 252)$=3.03, p<0.001. Aphantasics showed reduced activation of bilateral hippocampi, including the left anterior (p=0.033), left posterior (p=0.027), right anterior (p=0.047), and right posterior hippocampus (p=0.025). There was no laterality effect nor differences along the pattern of activation down the anterior-posterior axis between the groups (all p>0.05). These findings indicate that the behavioral AM deficit associated with aphantasia is reflected neurally by a reduced bilateral hippocampal activation.

### Activation patterns associated with AM retrieval

Second, we examined whole-brain activation during AM retrieval of both groups and the results are displayed in *Figure 3a and b*. Additionally, the peak coordinates of AM and MA activation for aphantasics and controls are shown in *Tables 1 and 2*, respectively.

Overall, both groups showed greater activation in all areas typically associated with AM, including bilateral hippocampus, vmPFC, and medial/lateral parietal regions, during AM retrieval. When examining the group differences, aphantasics displayed greater activation in bilateral visual-perceptual regions (maximum in lingual gyrus) in the occipital lobe than controls, $t(28)$ = 4.41, p<0.001 (MNI: right visual cortex: x=12, y = −79, z=5; left visual cortex: x = −9, y = −76, z=29, see *Figure 3c, d and e*). In contrast, controls showed greater activation in the right hippocampus than aphantasics, $t(28)$ = 3.77, p<0.001 (MNI: x=39, y = −31, z = −13). An additional correlational analysis revealed that those participants with higher visual-perceptual cortex activation had less hippocampal activation, $r(28)$ = −0.39,

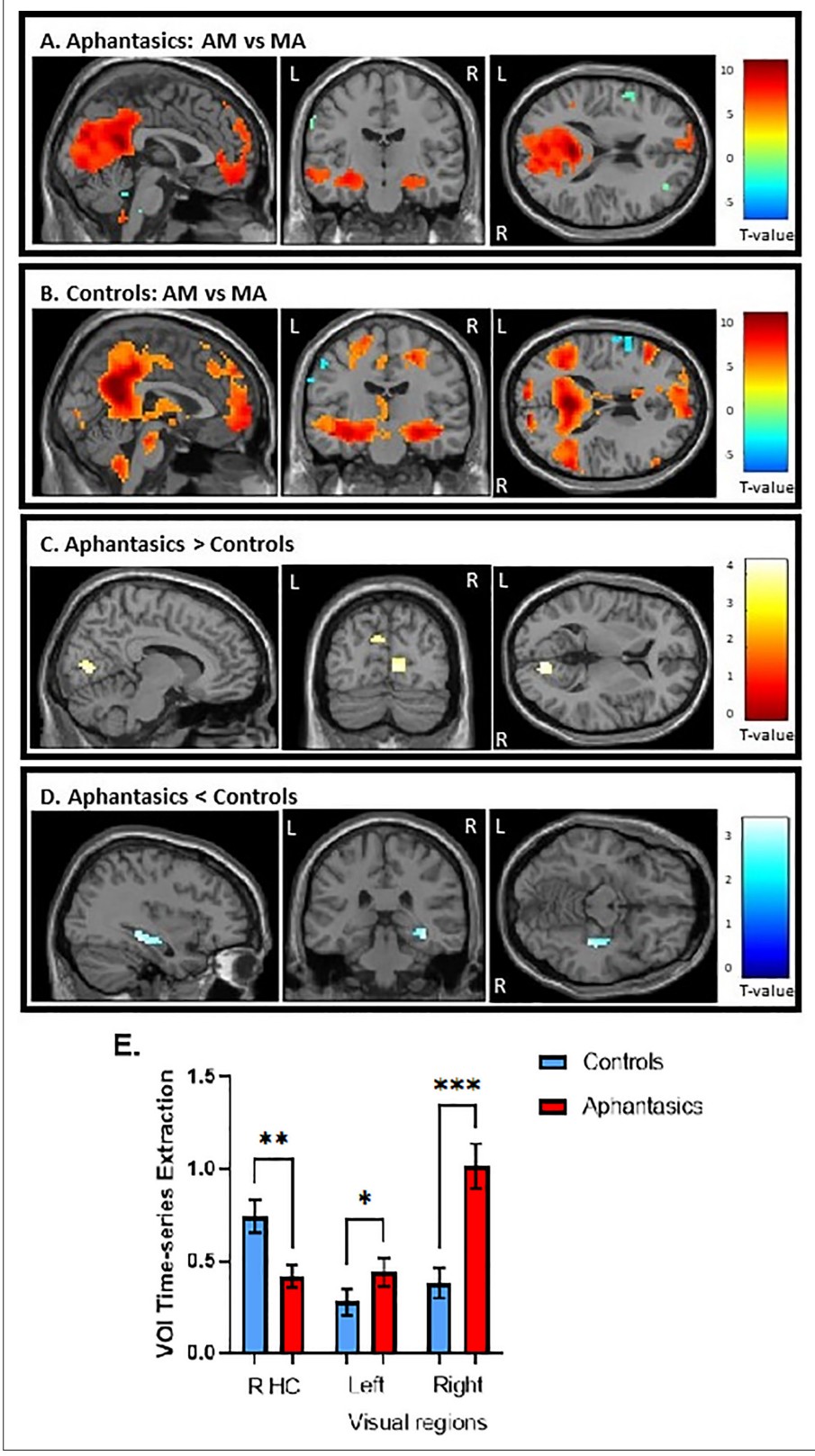

**Figure 3.** Activation during the autobiographical memory retrieval task. (**A**) Stronger activated cortical regions during AM retrieval (in warm colors) in comparison to mental arithmetic (in cool colors) in aphantasics and (**B**) controls. (**C**) Aphantasics showed greater activation in visual-perceptual cortices than controls, and (**D**) controls showed stronger activation in the right posterior hippocampus than aphantasics. Images are thresholded at

*Figure 3 continued on next page*

*Figure 3 continued*

p<0.001, cluster size 10, uncorrected, except (**D**) which is thresholded at p<0.01, cluster size 10, for display purposes only (i.e. the peak voxel and adjacent 10 voxels also survived p<0.001, uncorrected). (**E**) The percentage of signal change for the contrast AM versus MA were extracted from the peaks of activated voxels, each with 1 mm sphere for display purposes.

**Table 1.** Peak coordinates of the AM and MA activation for Aphantasia.

| Region | Hemisphere | MNI Coordinates | | | Voxels | T-value |
|---|---|---|---|---|---|---|
| | | X | Y | Z | | |
| **Activation AM >MA** | | | | | | |
| Posterior Cingulate Gyrus | Right | 18 | –57 | 11 | 4657 | 11.00 |
| Parahippocampual Gyrus* | Left | –21 | –31 | –13 | | 9.06 |
| Hippocampus | Left | –27 | –17 | –19 | 205 | 8.40 |
| Superior Frontal Gyrus | Left | –12 | 47 | 50 | 926 | 8.38 |
| Angular Gyrus | Left | –42 | –55 | 23 | 165 | 7.88 |
| Lateral Orbitofrontal Cortex | Left | –42 | 38 | –16 | 208 | 7.58 |
| Hippocampus | Right | 18 | –37 | -1 | 199 | 6.89 |
| Cerebellum | Right | 15 | –79 | –37 | 109 | 6.33 |
| Brainstem | Right | 3 | –46 | –52 | 43 | 6.03 |
| Parahippocampual Gyrus* | Right | 24 | –31 | –13 | | 6.02 |
| Middle Temporal Gyrus | Right | 60 | 2 | -–19 | 76 | 5.27 |
| Supramarginal Gyrus | Right | 54 | –58 | 32 | 26 | 4.99 |
| Middle Frontal Gyrus | Left | –39 | 20 | 50 | 12 | 4.52 |
| **Activation MA >AM** | | | | | | |
| Precuneus | Left | –18 | –58 | 41 | 594 | –3.85 |
| Inferior Temporal Gyrus | Right | 51 | –46 | –13 | 123 | –3.85 |
| Precuneus | Right | 24 | –49 | 53 | 718 | –3.85 |
| Insula | Left | –30 | 23 | 11 | 48 | –3.85 |
| Inferior Temporal Gyrus | Left | –51 | –49 | –13 | 67 | –3.86 |
| Cerebellum | Right | 30 | –67 | –52 | 27 | –3.87 |
| Middle Frontal Gyrus | Right | 33 | 41 | 17 | 34 | –3.87 |
| Superior Frontal Gyrus | Right | 30 | 5 | 59 | 52 | –3.87 |
| Inferior Frontal Gyrus | Right | 54 | 14 | 29 | 35 | –3.88 |
| Insula | Right | 39 | 11 | 8 | 51 | –3.88 |
| Inferior Frontal Gyrus | Left | –57 | 11 | 26 | 182 | –3.88 |
| Lateral Globus Pallidus | Right | 23 | -7 | 14 | 14 | –3.92 |
| Cerebellum | Left | –24 | –64 | –46 | 16 | –3.93 |

*Sub-cluster level, Cluster size = 10 voxels, p-value = 0.001.

**Table 2.** Peak coordinates of the AM and MA activation for healthy controls.

| Region | Hemisphere | MNI Coordinates | | | Voxels | T-value |
|---|---|---|---|---|---|---|
| | | X | Y | Z | | |
| **Activation AM >MA** | | | | | | |
| Parahippocampal Gyrus | Right | 27 | –28 | –19 | 11319 | 12.41 |
| Parahippocampal Gyrus* | Left | –24 | –25 | –16 | | 9.01 |
| Cerebellum | Left | –18 | –76 | –37 | 108 | 7.67 |
| Anterior Cingulate | Right | 9 | 35 | 11 | 13 | 7.01 |
| Medial Frontal Gyrus | Right | 18 | 32 | 29 | 233 | 6.93 |
| Inferior Frontal Gyrus | Right | 60 | 32 | 11 | 53 | 5.94 |
| Hippocampus | Left | –36 | –22 | –16 | 252 | 5.64 |
| Hippocampus | Right | 27 | –22 | –16 | 233 | 5.28 |
| Hypothalamus | Right | 3 | -4 | –10 | 16 | 4.93 |
| **Activation MA >AM** | | | | | | |
| Post Central Gyrus | Left | –33 | –43 | 62 | 643 | –3.73 |
| Precuneus | Right | 21 | –52 | 53 | 483 | –3.74 |
| Inferior Frontal Gyrus | Right | 51 | 8 | 26 | 16 | –3.74 |
| Middle Occipital Gyrus | Right | 33 | –82 | 2 | 26 | –3.75 |
| Middle Temporal Gyrus | Left | –51 | –58 | -1 | 18 | –3.76 |

*Sub-cluster level, Cluster size = 10 voxels, p-value = 0.001.

p=0.041, indicating that there was a trade-off between increased visual-perceptual cortex activation and decreased hippocampal activation.

## Exploring functional connectivity of hippocampus and visual-perceptual cortices during AM

The whole-brain analyses strengthened our hypothesis that a core difference between aphantasics and controls lies in the interplay between the visual-perceptual cortex and the hippocampus. To test this interplay, in a third step, we examined functional connectivity between the peak differences of the hippocampus and the visual-perceptual cortex during AM retrieval (see *Figure 4*). We found a group difference in functional connectivity between the right hippocampus and left visual-perceptual cortices, *t*(28) = 2.65, p=0.006. Interestingly, while aphantasics show almost no functional connectivity

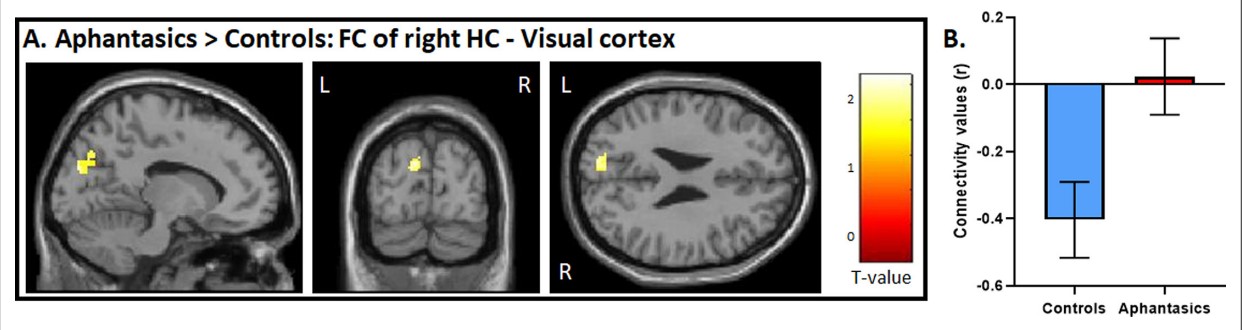

**Figure 4.** Functional connectivity between the visual-perceptual cortex and hippocampus during AM retrieval. (**A**) During AM retrieval, group differences in functional connectivity amongst the ROIs were only found between the right hippocampus, and left visual-perceptual cortices. (**B**) Controls displayed a stark negative correlation, whereas aphantasics did not. Image is displayed at p<0.05, small volume corrected, and a voxel cluster threshold of 10 adjacent voxels.

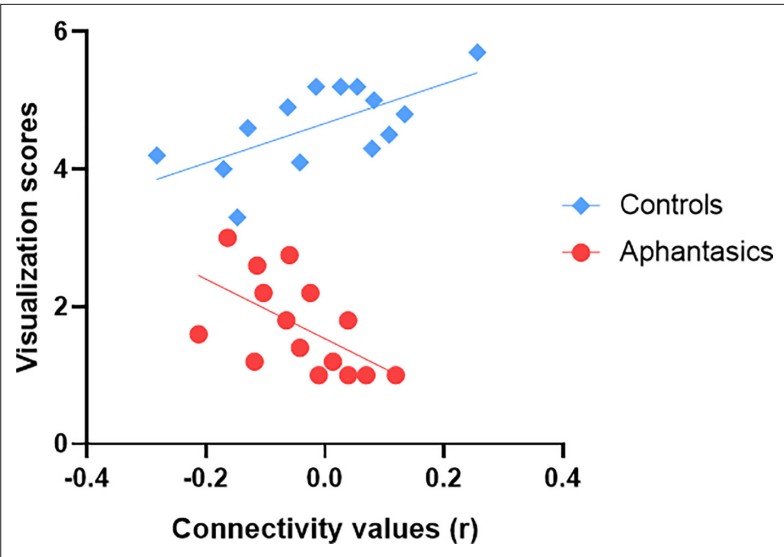

**Figure 5.** Functional connectivity between the visual-perceptual cortex and hippocampus during resting-state explains visualization abilities. Resting-state functional connectivity between the right hippocampus and the right visual-perceptual cortex correlates with visualization abilities. Fitted straight lines indicate a negative correlation for aphantasics (red) and a positive correlation for controls (blue).

between those two ROIs, controls displayed a strong negative connectivity between the two locations, p=0.013.

## Exploring functional connectivity of hippocampus and visual-perceptual cortices during resting-state

Lastly, we examined resting-state connectivity between our identified ROIs in the right posterior hippocampus and left and right visual-perceptual cortex. There was no group difference during resting-state between these ROIs. In order to examine whether resting-state functional connectivity carried information about one's ability to visualize AMs, we added the visualization scores as a regressor of interest in our model. While connectivity alone did not predict the visualization scores in the interview, $F(14, 40)=1.651$, p=0.391, β = –0.06, we found a main effect of group, $F(3, 40)=353.2$, p<0.001, $β=0.92$, and an interaction between group and connectivity, $F(3, 54)=305.1$, p<0.001, $β=0.26$. Interestingly, for controls, we found a positive correlation between the resting-state connectivity of the right hippocampus and the visual cortex and the visualization scores from the interview, $r(13) = 0.65$, p=0.011 (see *Figure 5*). On the other hand, for aphantasics, we found a negative correlation between the resting-state connectivity of the right hippocampus and the visual cortex and the visualization scores from the interview, $r(14) = –0.57$, p=0.027.

In sum, our fMRI results indicate that the impaired AM retrieval associated with aphantasia is reflected by functional alterations of the hippocampus and visual-perceptual cortex, as well as the interaction between them.

## Discussion

In this study, we set out to examine the neural correlates of episodic AM retrieval in aphantasia as a way to examine the influence of visual imagery on episodic AM. In line with previous reports, we found that aphantasics reported less sensory details during AM retrieval regardless of the recency of memory (*Dawes et al., 2020*; *Dawes et al., 2022*; *Zeman et al., 2020*; *Zeman et al., 2020*). Strikingly, the deficit in constructing visual imagery associated with aphantasia did not only lead to a reduced retrieval of visual-perceptual details but to a broader impairment in retrieving episodic AMs, including reduced emotions and confidence attached to the memories. Thus, in agreement with a recent account of aphantasia (*Blomkvist, 2023*), our results support the idea that a diminished construction of visual details during AM retrieval leads to a more general episodic memory deficit.

We expand the current knowledge by adding that this AM deficit is reflected neurally by an altered activation and connectivity pattern between the hippocampus and visual-perceptual cortices. Our findings provide novel insights into three current debates: (1) the mechanisms of aphantasia-related AM deficits, (2) the similarities and differences between aphantasics and individuals with hippocampal damage, and (3) the neural models of AM.

## Potential mechanisms underlying aphantasia-related AM deficits

We report that aphantasics show increased activation of bilateral visual-perceptual cortices as well as decreased hippocampal activation during AM retrieval in comparison to controls. Increased activity in the visual-perceptual cortices in aphantasics has been reported previously, albeit not associated with AM (*Fulford et al., 2018*; *Keogh et al., 2020*). In a prominent review, Pearson synthesizes evidence about the neural mechanism of imagery strength (*Pearson, 2019*). Indeed, activity metrics in the visual cortex predict imagery strength (*Cui et al., 2007*; *Dijkstra et al., 2017*). Interestingly, lower resting activity and excitability result in stronger imagery, and reducing cortical activity in the visual cortex via transcranial direct current stimulation (tDCS) increases visual imagery strength (*Keogh et al., 2020*). Thus, one potential mechanism of aphantasia-related AM deficits is that the heightened activity of the visual-perceptual cortices observed in our and previous work hinders aphantasics to detect weaker imagery-related signals.

Further, we had the a priori hypothesis that hippocampal activation will be decreased in aphantasics; a hypothesis which we could confirm in our native space hippocampal analysis. On the whole-brain level, perhaps due to our small sample size, only the cluster of activation group differences in the right posterior hippocampus survived the statistical threshold. Given the low power, further studies are needed to confirm this effect; however, the right posterior hippocampus interacts in the healthy brain heavily with the visual-perceptual cortex only during the elaboration phase of AM retrieval (*McCormick et al., 2015*).

In addition, controls exhibited a strong functional connection between both brain structures during AM retrieval and this functional connectivity predicted better visualization skills. At first glance, it is surprising that this functional connectivity was negative. However, negative visual-perceptual cortex activation during perceiving and imagining scenes has been reported before (*McCormick et al., 2020*). One possible explanation might be that signals from the hippocampus selectively inhibit imagery-irrelevant activation in the visual-perceptual cortices (e.g. sensory noise) to carve out imagery-related signals (*Pace et al., 2023*). This would be in line with the hypothesis stated above, that a bad signal-to-noise ratio in the visual cortex hiders aphantasics to create mental imagery. Either way, the described functional connectivity between the hippocampus and the visual-perceptual cortex fits well to previous neuroimaging studies pointing towards a central role of the dynamic interplay between these brain structures during AM retrieval (*McCormick et al., 2015*). This interplay seems to be especially important during the elaboration stage of AM retrieval, a period when specific visual-perceptual details are being actively brought back into the mind's eye. At this point, however, it remains unclear whether the disruption of AM elaboration associated with aphantasia takes place during the encoding, storage, or retrieval process.

From a theoretical point of view, the extended constructive episodic simulation hypothesis proposes a top-down hierarchy during mental imagery (*Blomkvist, 2023*). In this model, the hippocampus initiates retrieval processes in primary sensory brain regions, such as the visual-perceptual cortex in order to retrieve visual-perceptual details associated with a specific AM. Evidence for such top-down hierarchies during mental imagery have been observed in fronto-parietal and occipital networks via effective connectivity analyses, such as Granger Causality and Dynamic Causal Modelling (*Dentico et al., 2014*; *Dijkstra et al., 2017*; *Mechelli et al., 2004*). For example, intracranial and high-density scalp electroencephalography (EEG) provided evidence of high-frequency hippocampal signaling during the recall of perceptual cues in patients with epilepsy, indicating that the hippocampus drives the switch from perception to memory recalling (*Treder et al., 2021*). In aphantasia, it is hypothesized that this top-down hierarchy is disrupted and therefore, the hippocampus can no longer initiate the retrieval and incorporation of visual-perceptual details in one coherent mental event. Because of the slow temporal resolution of our fMRI sequence, our data cannot directly speak to the question of temporal directionality between the hippocampus and visual-perceptual cortex. Nonetheless, our findings suggest that the bidirectional connectivity between both brain structures is crucial for the

re-experience of episodic AMs. As such, hippocampal processes may be needed to retrieve specific details and if these details are not provided by the visual-perceptual cortices, the entire episodic AM retrieval seems to fail.

## Similarities and differences between aphantasics and individuals with hippocampal lesions

At face value, the episodic AM deficits in aphantasia in our data and reported previously (*Dawes et al., 2020*; *Dawes et al., 2022*; *Zeman et al., 2020*; *Zeman et al., 2020*), as well as the decreased hippocampal activation during AM retrieval suggest that aphantasia is a selective episodic memory condition (see *Blomkvist, 2023*), similar to AM amnesia known from individuals with hippocampal damage. In fact, previous and the current study show that aphantasics and individuals with hippocampal damage report less internal details across several memory detail subcategories, such as emotional details and temporal details (*Rosenbaum et al., 2008*; *St Laurent et al., 2009*; *Steinvorth et al., 2005*), and these deficits can be observed regardless of the recency of the memory (*Miller et al., 2020*). These similarities suggest that aphantasics are not merely missing the visual-perceptual details to specific AM, but they have a profound deficit associated with the retrieval of AM.

Nonetheless, there are also stark differences between aphantasics and individuals with hippocampal damage. Foremost, aphantasics seem not to have difficulties to retrieve spatial information (*Bainbridge et al., 2021*), which is another inherent function of the hippocampus (*Burgess et al., 2002*; *O'Keefe, 1991*). In the current study, we did not set out to examine spatial cognition in aphantasics; however, parts of our data speak to this aspect. While in our study aphantasics reported less amount of spatial details during the AI, this standard scoring procedure only counts place details when the exact place is recalled and is not meant to assess the recall of spatial layout (*Levine et al., 2002*). Thus, this place score may not represent spatial cognition per se. In fact, when asking aphantasics about their experience, they point out difficulties in recalling AM and using imagination in daily life, however they report no difficulties in spatial orientation. Indeed, often during the interview, aphantasics would explain that they know how the space around them felt, they just cannot see it in front of their mind's eye. One aphantasic put her finger on it, describing it as: "I can put my consciousness in my kitchen at home and feel all around but there is no visual image attached to this feeling." These observations support the idea that some hippocampal processes, at least regarding spatial navigation, may be intact in people with aphantasia (*Bainbridge et al., 2021*). However, spatial cognition should be formally addressed in future studies. One way to assess this hippocampal function would be to examine tasks, which rely on scene construction. The scene construction theory states that the hippocampus is crucially needed for the construction of spatially coherent mental models of scenes (*Maguire and Mullally, 2013*). For example, patients with hippocampal damage cannot imagine the spatial layout of fictitious scenes (*Hassabis et al., 2007*), they detect less errors in spatially incoherent scenes than controls (*McCormick et al., 2017*), and they show less scene-dependent mind-wandering episodes (*McCormick et al., 2018b*). In contrast, we would predict that aphantasics have diminutive deficits in tasks that depend on hippocampal scene construction processes.

What could be impaired in aphantasics are all cognitive functions which rely on the population of the constructed scenes with visual-perceptual details, such as episodic AM retrieval, episodic future thinking, complex decision-making, and complex empathy tasks.

## Towards a novel neural model of autobiographical memory

While more research is required exploring the cognitive landscape associated with aphantasia, such as spatial cognition and scene construction, our data contribute to an old debate of how AM retrieval and visual imagery are intertwined. We propose that the hippocampus is embedded in a brain-wide network, comprising the vmPFC and visual-perceptual cortices, in which each of these nodes contributes specific processes to the re-construction of extended detail-rich mental events (see also *Ciaramelli et al., 2019*; *McCormick et al., 2018a*). Within this model, the vmPFC initiates and oversees the scene construction process which takes place in the hippocampus. Further, the visual-perceptual cortex provides the visual details which are essential to populate the hippocampally constructed scenes. This model is backed up by a previous MEG study revealing that the vmPFC directs hippocampal activity during the initiation of AM retrieval (*McCormick et al., 2020*). This finding has been replicated and extended by *Chen et al., 2021*, showing that the vmPFC leads hippocampal involvement during

scene construction and other scene-based processes (*Monk et al., 2021*). Moreover, the connection between the hippocampus and the visual-perceptual cortex seems equally crucial. There are a few case reports of damage to the occipital cortex causing AM amnesia (*Greenberg et al., 2005*), potentially by preventing the population of the hippocampally constructed scenes. Furthermore, our current study suggests that a reliable connectivity between the hippocampus and the visual-perceptual cortices is important to provide the visual details necessary for successful vivid, detail-rich AM retrieval.

## Conclusion

Aphantasia provides a natural knock-out model for the influence of visual imagery on different cognitive functions. We here report a tight link between visual imagery and our ability to retrieve vivid and detail-rich personal past events, as aphantasics do not only report fewer visual-perceptual details during episodic AM retrieval but also show decreased confidence and emotionality associated with these memories. In this context, we highlight the central role of the functional connectivity between the hippocampus and occipital cortex to assemble visual-perceptual details into one coherent extended mental event. Exciting novel research avenues will be to examine hippocampal-dependent spatial cognition in aphantasics and to investigate whether neuroscientific interventions can be used to enhance AM retrieval by enhancing visual imagery.

# Materials and methods

## Participants

In total, 31 healthy individuals with no previous psychiatric or neurological condition participated in this study. Fifteen congenital aphantasics and 16 matched controls were recruited from the database of the *Aphantasia Research Project Bonn* (*Monzel et al., 2021*; *Monzel et al., 2022b*). Due to technical issues during MRI scanning, one participant (with aphantasia) had to be excluded from the analyses. Groups were matched for basic demographic data, that is, sex, age, and education, as well as intelligence assessed with a short intelligence screening (*Baudson and Preckel, 2015*; see *Table 3*). Oral and written informed consent was obtained from all participants prior to the commencement of

**Table 3.** Demographic data for aphantasics, controls and the total sample.

|  | Total (n=30) | Aphantasics (n=14) | Controls (n=16) | Test statistic | p | $BF_{01}$ |
|---|---|---|---|---|---|---|
| Age |  |  |  | 0.80* | .431 | 2.30 |
| *M* | 29.77 | 31.47 | 28.19 |  |  |  |
| *SD* | 11.36 | 10.45 | 12.27 |  |  |  |
| IQ |  |  |  |  |  |  |
| *M* | 93.77 | 91.73 | 95.69 | 0.81* | .425 | 2.29 |
| *SD* | 13.53 | 16.61 | 10.02 |  |  |  |
| Sex |  |  |  | 2.76† | .097 | 0.69 |
| Male (%) | 32.3 | 53.3 | 81.3 |  |  |  |
| Female (%) | 67.7 | 46.7 | 18.8 |  |  |  |
| Education |  |  |  | 1.59† | .662 | 7.90 |
| Secondary school (%) | 6.5 | 6.7 | 6.3 |  |  |  |
| A-levels (%) | 35.5 | 40.0 | 31.3 |  |  |  |
| University degree (%) | 54.8 | 46.7 | 62.5 |  |  |  |
| Doctoral degree (%) | 3.2 | 6.7 | 0.0 |  |  |  |

Note. $BF_{01}$=Bayes Factor, indicates how much more likely H0 is compared to H1.

*t-test.

† $\chi^2$-test.

experimental procedure in accordance with the Declaration of Helsinki (*World Medical Association, 2013*) and the local ethics board of the University Hospital Bonn.

## Vividness of visual imagery questionnaire

Aphantasia is typically assessed with the Vividness of Visual Imagery Questionnaire (VVIQ; *Marks, 1973*; *Marks, 1995*), a subjective self-report questionnaire that measures how vivid mental scenes can be constructed by an individual. For example, individuals are asked to visualize a sunset with as much details as possible and rate their mental scene based on a 5-point Likert scale (ranging from 'no image at all, you only "know" that you are thinking of the object' to 'perfectly clear and as vivid as normal vision'). Since there are 16 items, the highest score of the VVIQ is 80 indicating the ability to visualize mental images with such vividness as if the event were happening right there and then. The minimum number of points is 16 indicating that an individual reported no mental image for any of the items at all. Aphantasia is at the lower end of the spectrum of imagery-abilities and usually identified with a VVIQ-score between 16 and 32 (e.g. *Dawes et al., 2020*; *Dawes et al., 2022*).

## Binocular rivalry task

Since self-report questionnaires such as the VVIQ are associated with several drawbacks, such as their reliance on introspection (*Schwitzgebel, 2002*), we administered a mental imagery priming-based binocular rivalry task to assess mental imagery more objectively (for more details, see *Keogh and Pearson, 2018*; *Pearson et al., 2008*). In short, after imagining either red-horizontal or blue-vertical Gabor patterns, participants were presented with a red-horizontal Gabor pattern to one eye and a blue-vertical Gabor pattern to the other eye. Subsequently, participants were asked to indicate which type of Gabor pattern they predominantly observed. Usually, successful mental imagery leads subjects to select the Gabor pattern which they had just visualized. This selection bias can be transferred into a priming score representing visual imagery strength. (Imagery strength is used to describe the results of the Binocular Rivalry Task, whereas vividness of mental imagery is used to describe the results of the VVIQ. Although both tasks are correlated, the VVIQ measures vividness, whereas the dimension of the Binocular Rivalry Task is not clearly defined.) Mock stimuli consisting of only red-horizontal or blue-vertical Gabor patterns were displayed in 12.5% of the trials to be able to detect decisional biases. Previous studies have shown that the binocular rivalry task validly correlated with mental imagery strength (*Pearson et al., 2011*; *Wagner and Monzel, 2023*).

## Autobiographical interview

Detailed behavioral AM measures were obtained in blinded semi-structured interviews either in-person or online via Zoom (Zoom Video Communications Inc, 2016) using the Autobiographical Interview (AI; *Levine et al., 2002*). All interviews were conducted in German. During the AI, the interviewer asks the participant to recall five episodic AMs from different life periods: early childhood (up to age 11), adolescent years (ages 11–17), early adulthood (ages 18–35), middle age (35–55), and the previous year. In order to acquire five AMs in every participant, the middle age memory was replaced by another early adulthood memory for participants who were younger than 34 years old (see *Levine et al., 2002*). Hence, all participants provided the last time period with memories from their previous year. Memories from the first four periods were considered remote, whereas the memory from the previous year was considered recent. The interview is structured so that each memory recollection consists of three parts: free recall, general probe, and specific probe. During free recall, the participants were asked to recall as many details as possible for a memory of their choice that is specific in time and place within the given time period. When the participant came to a natural ending, the free recall was followed by the general and specific probes. During the general probe, the interviewer asked the participant encouragingly to provide any additional details. During the specific probe, specific questions were asked for details about the time, place, perception, and emotion/thoughts of each memory. Then, participants were instructed to rate their recall in terms of their ability to visualize the event on a 6-point Likert scale (ranging from 'not at all' to 'very much'). The interview was audiotaped, and afterwards transcribed and then scored by two independent raters according to the standard protocol (*Levine et al., 2002*). The interviews were scored after all data had been collected, in random order, and scorers were blind to the group membership of the participant.

For scoring, the memory details were assigned to two broad categories, that is, internal and external details. There were the following subcategories of internal details: internal events (happenings, weather conditions, people present, actions), place (country, city, building, part of room), time (year, month, day, time of the day), perceptual details (visual, auditory, gustatory, tactile, smell, body position), and emotion/thought (emotional state, thoughts). The subcategories for external details were semantic details (factual or general knowledge), external events (other specific events in time and place but different to the main event), repetition (repeated identical information without request), and other details (metacognitive statements, editorializing). In addition, following the standard procedure, an episodic richness score was given for each memory by the rater on a 7-point Likert scale (ranging from 'not at all' to 'perfect'). Furthermore, we added a novel rating score of confidence to the protocol since many participants indicated very strong belief in the details they provided, while others were insecure about the correctness of their own memories. Confidence scores were again rated on a 7-point Likert scale (ranging from 'not at all' to 'perfect').

## Debriefing questions

Following the AI, we asked participants three general questions. These were thought of as open questions to get people to talk about their personal perspective on AM, spatial cognition, and imagination.

1. Typically, how difficult is it for you to recall autobiographical memories?
2. Typically, how difficult is it for you to orient yourself spatially?
3. Typically, how difficult is it for you to use your imagination?

After a free report, participants were asked to rate the difficulty on a Likert-scale from 1 ('very easy') to 6 ('very difficult').

## Autobiographical memory fMRI task

The experimental fMRI task was adapted from a previous protocol by *McCormick et al., 2015*. Two conditions, an AM retrieval task and a simple math task (MA), each consisting of 20 randomized trials, were included in this experiment. During AM trials, cue words, such as 'a party', were presented on the screen for 12 s and participants were instructed to recall a personal event related to the word cue which was specific in time and place (e.g. their 20th birthday party). Participants were asked to press a response button once an AM was retrieved to indicate the time point by which they would start to engage in the AM elaboration phase. For the rest of the trial duration, participants were asked to re-experience the chosen AM and try to recall as many details as possible without speaking out loud. After each AM trial, participants were instructed to rate via button presses whether their retrieval had been vivid or faint. We chose a simple two-button response in order to keep the task as easy as possible. During MA trials, simple addition or subtraction problems, for example, 47+19, were presented on the screen for 12 s. Here, participants were instructed to press a response button once the problems were solved and asked to engage in adding 3 s to the solutions, for example, (47+19)+3 + ...+3, until the trial ended. The MA trials were followed by a rating whether the MA problems had been easy or difficult to solve. For both AM and MA, each trial lasted for 12 s, the maximum time for rating of 3 s, and a jittered inter-stimulus interval (ISI) between 1–4 s. Since we were especially interested in the elaboration phase of AM retrieval, for the fMRI analyses, we modelled the last 8 s of each AM and MA trial just before the rating screen appeared.

## MRI data acquisition

Anatomical and functional data were acquired at the German Center for Neurodegenerative Diseases (DZNE), Bonn, Germany, using a 3 Tesla MAGNETOM Skyra MRI scanner (Siemens Healthineers, Erlangen, Germany). A mirror was mounted on the 32 channel receiver head coil and was placed in the scanner for the participants to view the stimuli shown on an MRI conditional 30-inch TFT monitor (Medres medical research, Cologne, Germany) placed at the scanner's rear end. The MRI protocol consisted of anatomical, resting-state, and AM task-based fMRI scanning sessions. In addition, we acquired further experimental fMRI and DTI data which are not part of this manuscript. Of note, the resting-state scans were acquired before participants engaged in the AM task in order to prevent reminiscing about personal memories during the resting-state. For the anatomical scans, an in-house developed 0.8 mm isotropic whole-brain T1-weighted sagittal oriented multi-echo

magnetization prepared rapid acquisition gradient echo (MEMPRAGE; *Brenner et al., 2014*) was employed with the following parameters: TR = 2.56 s, TEs = 1.72/3.44/5.16/6.88ms, TA = 6:48, matrix = 320 x 320 x 224, readout pixel bandwidth=680 Hz/Pixel, CAIPIRINHA mode. Resting-state (190 volumes, TA = 7 min) and AM task-based fMRI scans (460 volumes, TA = 15 min) were acquired using an interleaved multi-slice 3.5 mm isotropic echo planar imaging (EPI) sequence with TR = 2 s, TE = 30ms, matrix = 64 x 64 x 39, readout pixel bandwidth = 2112 Hz/Pixel (see *Jessen et al., 2018*). The images were obtained in an oblique-axial slice orientation along the anterior-posterior commissure line. During resting-state, the participants were asked to close their eyes and to think about nothing at all. The first 5 frames of each functional session were excluded for the scanner to reach equilibrium. Before each functional session, an optimal B0 shim was determined and individually mapped by 2-echo gradient echo (GRE) with same voxel resolution and positioning for later post-processing.

## Manual segmentation of the hippocampus

Since our main goal was to assess hippocampal involvement during AM retrieval in aphantasia, we sought to examine in depth whether there were any group differences in hippocampal activation in respect to the hemispheric laterality or along its long-axis. For this reason, we segmented the hippocampus based on the T1 structural images using ITKSnap (https://www.itksnap.org, Version 3.8). Although we did not segment specific hippocampal subfields, our masks included the dentate gyrus, CA1-4, subiculum and pre- and parasubiculum. Whole masks of the left and right hippocampus were segmented manually in their respective native space and were divided afterwards into anterior and posterior portions, using the location of the uncus as boundary.

## fMRI preprocessing

SPM12 (Statistical Parametric Mapping 12) software package (https://www.fil.ion.ucl.ac.uk/spm/) on MATLAB v19a (MathWorks) computing platform (https://matlab.mathworks.com/) was used to perform resting-state and AM task-based fMRI data preprocessing. The anatomical T1w RMS of all MEMPRAGE's echoes and functional 2D-EPI images were reoriented along the anterior-posterior commissure axis. The phase and magnitude images within the field maps were applied to calculate the voxel displacement maps (VDM) for geometrical correction of the distorted EPI images. The echo times were set to 4.92ms (short) and 7.38ms (long). The total EPI readout time was 34.56ms. The calculated EPI and VDMs were applied to the functional scans for realignment and unwarping. The functional scans were then co-registered to the segmented bias corrected T1 scans.

Whole-brain differences between groups were evaluated. Thus, co-registered scans were normalized to the Montreal Neurological Institute (MNI) space and a Gaussian smoothing kernel of 8 mm FWHM was applied. In addition, for functional connectivity analyses, denoising was applied using a linear regression model of potential confounding effects (global white matter signal, CSF signal, and ART scrubbing parameters) in the BOLD signal using CONN software package v20.b (https://www.nitric.org/projects/conn/). Temporal band pass filter was set from 0.01 to infinite to further minimize artifacts.

## Statistical analyses

### Behavioral analyses

Two samples t-tests were calculated to assess differences in the priming scores of aphantasics and controls in the binocular rivalry task. One sample t-tests were used to distinguish the performances of both groups from chance. To assess differences of Autobiographical Interview scores between aphantasics and controls, a 2x2 × 2 ANOVA with post-hoc t-tests were calculated with type of memory details (internal vs. external) and memory recency (remote vs. recent) as within-subject factors and group (aphantasics vs. controls) as between-subject factor. Afterwards, Bonferroni-corrected t-tests were conducted for specific internal (time, place, internal event, perception, emotion) and external (external event, semantic, repetition, other) memory details. Differences in memory ratings (confidence, episodic richness), self-reported visualization scores, debriefing questions, and behavioral responses during fMRI scanning were also assessed via two sample t-tests.

## Hippocampal activity associated with autobiographical memory in native space

In order to examine hippocampal activity associated with autobiographical memory, we extracted signal intensity values for both AM and MA trials for each participant for our manually segmented anatomical masks of the left and right, anterior and posterior portions of the hippocampus using the MATLAB-based Response Exploration toolbox (REX; https://www.nitric.org/projects/rex/). We then calculated for each participant for each anatomical mask the difference between AM and MA signal intensities. Afterwards, group differences between aphantasics and controls with respect to the laterality effects and effects between the anterior and posterior hippocampus were assessed using a two-way ANOVA with a post-hoc Tukey's multiple comparison test, applying a significance threshold of $\alpha=.05$.

## Whole-brain fMRI activation analyses

After focusing on the hippocampus, we examined group differences of whole-brain activation associated with AM and MA following the standard GLM procedure in SPM12. Owing to the prominence of mental imagery during the elaboration phase of AM retrieval, we analyzed the last 8 s of the AM and MA trials prior to the display of the vividness rating. These trials were modelled as mini blocks in the GLM with motion correction regressors included as covariate of no interest. We specified our main contrast of interest, that is AM versus MA on the first level, which was then brought to the second group level using a one-sample t-test. Finally, the activation maps of the two groups were compared using a two-sample t-test. For whole-brain analysis, we applied a significance threshold of $p<0.001$, and voxel cluster size of 10 adjacent voxels, uncorrected.

## ROI-to-ROI functional connectivity analyses

One of our main a priori hypotheses stated that the hippocampus and the visual-perceptual cortex show differential engagement during AM retrieval associated with aphantasia which was confirmed by our whole-brain activation analyses. Based on these results, we sought to examine the functional connectivity between those two areas. Towards this end, we created regions of interest (ROIs, spheres with a diameter of 10 mm consisting of 536 voxels) around the three peaks of the activation differences, following the whole-brain fMRI activation analyses, using the MarsBaR HBM toolbox (*Brett et al., 2002*). The ROIs comprised (1) the right hippocampus, MNI: x=39, y = –31, z = –13, (2) the right visual cortex, MNI: x=12, y = –79, z=5, and (3) the left visual cortex, MNI: x = –9, y = –76, z=29. Using CONN, we examined functional connectivity (i.e. Generalized Psycho-Physiological Interactions, weighted general linear model with bivariate correlation) between the hippocampal ROI and the ROIs situated in the visual-perceptual cortex during AM task-based fMRI and during resting-state. Furthermore, in order to examine how well functional connectivity between hippocampus and the visual cortex reflected an individuals' ability to visualize mental events, we examined a regression model with the visualization scores of the AI as criterion and resting state connectivity values, group allocation and the interaction term of the connectivity values and group allocation as predictors. For these a priori driven analyses, we applied a significance threshold of $p<0.05$, small volume corrected, and a voxel cluster threshold of 10 adjacent voxels.

## Acknowledgements

CM was supported by the German Research Foundation (DFG, MC244/3-1) and the DZNE Foundation (DZNE Stiftung-Forschung für ein Leben ohne Demenz, Parkinson & ALS). We thank Anke Rühling and Jennifer Schlee for their technical support during scanning and Yilmaz Sagik for serving as the secondary scorer for the Autobiographical Interview. In addition, we would like to thank all participants for their time and effort to share their memories with us.

# Additional information

## Funding

| Funder | Grant reference number | Author |
| --- | --- | --- |
| Deutsche Forschungsgemeinschaft | MC244/3-1 | Cornelia McCormick |
| Deutsches Zentrum für Neurodegenerative Erkrankungen | BN059K | Cornelia McCormick |

The funders had no role in study design, data collection and interpretation, or the decision to submit the work for publication.

## Author contributions

Merlin Monzel, Conceptualization, Data curation, Formal analysis, Investigation, Methodology, Writing – original draft, Writing – review and editing; Pitshaporn Leelaarporn, Data curation, Formal analysis, Visualization, Writing – original draft, Writing – review and editing; Teresa Lutz, Data curation, Formal analysis, Investigation; Johannes Schultz, Conceptualization, Supervision, Validation, Methodology, Writing – review and editing; Sascha Brunheim, Software, Supervision, Validation, Investigation, Methodology, Writing – review and editing; Martin Reuter, Conceptualization, Resources, Supervision, Validation, Methodology, Project administration, Writing – review and editing; Cornelia McCormick, Conceptualization, Resources, Data curation, Formal analysis, Supervision, Funding acquisition, Validation, Methodology, Project administration, Writing – review and editing

## Author ORCIDs

Merlin Monzel ![ORCID] https://orcid.org/0000-0001-7012-9350
Pitshaporn Leelaarporn ![ORCID] http://orcid.org/0000-0001-8755-875X
Sascha Brunheim ![ORCID] https://orcid.org/0000-0001-9341-797X

## Ethics

Oral and written informed consent was obtained from all participants prior to the commencement of the experimental procedure in accordance with the Declaration of Helsinki (World Medical Association, 2013) and the local ethics board of the University Hospital Bonn.

Reviewer #1 (Public Review): https://doi.org/10.7554/eLife.94916.3.sa1
Reviewer #2 (Public Review): https://doi.org/10.7554/eLife.94916.3.sa2
Author response https://doi.org/10.7554/eLife.94916.3.sa3

---

# Additional files

## Supplementary files
• MDAR checklist

## Data availability

We do not have ethical approval of our research institution and our participants to share the raw brain imaging data publicly. However, the processed data can be accessed via Dryad. We also hold the permission to share raw data with scientific collaborators (no commercial research), so that interested researchers may contact cornelia.mccormick@dzne.de. There is no need for a research proposal.

The following dataset was generated:

| Author(s) | Year | Dataset title | Dataset URL | Database and Identifier |
| --- | --- | --- | --- | --- |
| Monzel M, Leelaarporn P, Lutz T, Schultz J, Brunheim S, Reuter M, McCormick C | 2024 | Data from: Hippocampal-occipital connectivity reflects autobiographical memory deficits in aphantasia | https://doi.org/10.5061/dryad.fbg79cp48 | Dryad Digital Repository, 10.5061/dryad.fbg79cp48 |

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
