## [Editor Report · eLife assessment]

This **important** work substantially advances our understanding of episodic memory in individuals with aphantasia, and sheds light on the neural underpinnings of episodic memory and mental imagery. The evidence supporting the conclusions is **convincing**, including evidence from a well-established interview paradigm complemented with fMRI to assess neural activation during memory recall. The work will be of broad interest to memory researchers and mental imagery researchers alike.

---

## [Referee Report · Reviewer #1 (Public Review)]

Summary:

In this article, the authors investigate whether the connectivity of the hippocampus is altered in individuals with aphantasia ¬- people who have reduced mental imagery abilities and where some describe having no imagery, and others describe having vague and dim imagery. The study investigated this question using a fMRI paradigm, where 14 people with aphantasia and 14 controls were tested, and the researchers were particularly interested in the key regions of the hippocampus and the visual-perceptual cortices. Participants were interviewed using the Autobiographical Interview regarding their autobiographical memories (AMs), and internal and external details were scored. In addition, participants were queried on their perceived difficulty in recalling memories, imagining, and spatial navigation, and their confidence regarding autobiographical memories was also measured. Results showed that participants with aphantasia reported significantly fewer internal details (but not external details) compared to controls; that they had lower confidence in their AMs; and that they reported finding remembering and imagining in general more difficult than controls. Results from the fMRI section showed that people with aphantasia displayed decreased hippocampal and increased visual-perceptual cortex activation during AM retrieval compared to controls. In contrast, controls showed strong negative functional connectivity between hippocampus and the visual cortex. Moreover, resting state connectivity between the hippocampus and visual cortex predicted better visualisation skills. The authors conclude that their study provides evidence for the important role of visual imagery in detail-rich vivid AM, and that this function is supported by the connectivity between the hippocampus and visual cortex. This study extends previous findings of reduced episodic memory details in people with aphantasia, and enables us to start theorising about the neural underpinnings of this finding.

The data provided good support for the conclusion that the authors draw, namely that there is a 'tight link between visual imagery and our ability to retrieve vivid and detail-rich personal past events'. However, as the authors also point out, the exact nature of this relationship is difficult to infer from this study alone, as the slow temporal resolution of fMRI cannot establish the directionality between the hippocampus and the visual-perceptual cortex. This is an exciting future avenue to explore.

Strengths:

A great strength of this study is that it introduces a fMRI paradigm in addition to the autobiographical interview, paralleling work done on episodic memory in cognitive science (e.g. Addis and Schacter, 2007, https://doi.org/10.1016%2Fj.neuropsychologia.2006.10.016), which has examined episodic and semantic memory in relation to imagination (future simulation) in non-aphantasic participants as well as clinical populations. Future work could build on this study, and for example use the recombination paradigm (Addis et al. 2009, 10.1016/j.neuropsychologia.2008.10.026), which would shed further light on the ability of people with aphantasia to both remember and imagine events. Future work could also build on the interesting findings regarding spatial navigation, which together with previous findings in aphantasia (e.g. Bainbridge et al., 2021, https://doi.org/10.1016/j.cortex.2020.11.014) strongly suggests that spatial abilities in people with aphantasia are unaffected. This can shed further light on the different neural pathways of spatial and object memory in general. In general, this study opens up a multitude of new avenues to explore and is likely to have a great impact on the field of aphantasia research.

Weaknesses:

A weakness of the study is that some of the questions used are a bit vague, and no objective measure is used, which could have been more informative. For example, the spatial navigation question (reported as 'How difficult is it typically for you to orient you spatially?' could have been more nuanced to tap into whether participants relied mostly on cognitive maps likely supported by the hippocampus) or landmarks. It would also have been interesting to conduct a spatial navigation task, as participants do not necessarily have insight to their spatial navigation abilities (they could have been overconfident or underconfident in their abilities). Secondly, the question 'how difficult is it typically for you to use your imagination?' could also be more nuanced, as imagination is used in a variety of ways, and we only have reason to hypothesise that people with aphantasia might have difficulties in some cases (i.e. sensory imagination involving perceptual details). It is unlikely that people with aphantasia would have more difficulty than controls to use their imagination to imagine counterfactual situations and engage in counterfactual thought (de Brigard et al., 2013, https://doi.org/10.1016%2Fj.neuropsychologia.2013.01.015) due to its non-sensory nature, but the question used does not distinguish between these types of imagination. Again, this is a ripe area for future research. The general phrasing of 'how difficult is [x]' could also potentially bias participants towards more negative answers, something which ought to be controlled for in future research.

---

## [Referee Report · Reviewer #2 (Public Review)]

Summary:

This study investigates to what extent neural processing of autobiographical memory retrieval is altered in people who are unable to generate mental images ('aphantasia'). Self-report as well as objective measures were used to establish that the aphantasia group indeed had lower imagery vividness than the control group. The aphantasia group also reported fewer sensory and emotional details of autobiographical memories. In terms of brain activity, compared to controls, aphantasics had a reduction in activity in the hippocampus and an increase in the activity in visual cortex during autobiographical memory retrieval. For controls, these two regions were also functionally connected during autobiographical memory retrieval, which did not seem to be the case for aphantasics. Finally, resting-state connectivity between visual cortex and hippocampus was positively related to autobiographical vividness in the control group but negatively in the aphantasia group. The results are in line with the idea that aphantasia is caused by an increase in noise within the visual system combined with a decrease in top-down communication from the hippocampus.

Recent years have seen a lot of interest in the influence of aphantasia on other cognitive functions and one of the most consistent findings is deficits in autobiographical memory. This is one of the first studies to investigate the neural correlates underlying this difference, thereby substantially increasing our understanding of aphantasia and the relationship between mental imagery and autobiographical memory.

Strengths:

One of the major strengths of this study is the use of both self-report as well as objective measures to quantify imagery ability. Furthermore, the fMRI analyses are hypothesis-driven and reveal unambiguous results, with alterations in hippocampal and visual cortex processing seeming to underlie the deficits in autobiographical memory.

Weaknesses:

In terms of weaknesses, the control task, doing mathematical sums, also differs from the autobiographical memory task in aspects that are unrelated to imagery or memory, such as self-relevance and emotional salience, which makes it hard to conclude that the differences in activity are reflecting only the cognitive processes under investigation. However, given that the most important comparisons are between groups of participants, this does not diminish the main conclusions about aphantasia.

Overall, I believe that this is a timely and important contribution to the field and will inspire novel avenues for further investigation.

---

## [Author Response]

The following is the authors’ response to the original reviews.

**Public Reviews:**

**Reviewer #1 (Public Review):**
Summary:In this article, the authors investigate whether the connectivity of the hippocampus is altered in individuals with aphantasia ¬- people who have reduced mental imagery abilities and where some describe having no imagery, and others describe having vague and dim imagery. The study investigated this question using a fMRI paradigm, where 14 people with aphantasia and 14 controls were tested, and the researchers were particularly interested in the key regions of the hippocampus and the visual-perceptual cortices. Participants were interviewed using the Autobiographical Interview regarding their autobiographical memories (AMs), and internal and external details were scored. In addition, participants were queried on their perceived difficulty in recalling memories, imagining, and spatial navigation, and their confidence regarding autobiographical memories was also measured. Results showed that participants with aphantasia reported significantly fewer internal details (but not external details) compared to controls; that they had lower confidence in their AMs; and that they reported finding remembering and imagining in general more difficult than controls. Results from the fMRI section showed that people with aphantasia displayed decreased hippocampal and increased visual-perceptual cortex activation during AM retrieval compared to controls. In contrast, controls showed strong negative functional connectivity between the hippocampus and the visual cortex. Moreover, resting state connectivity between the hippocampus and visual cortex predicted better visualisation skills. The authors conclude that their study provides evidence for the important role of visual imagery in detail-rich vivid AM, and that this function is supported by the connectivity between the hippocampus and visual cortex. This study extends previous findings of reduced episodic memory details in people with aphantasia, and enables us to start theorising about the neural underpinnings of this finding.The data provided good support for the conclusion that the authors draw, namely that there is a 'tight link between visual imagery and our ability to retrieve vivid and detail-rich personal past events'. However, as the authors also point out, the exact nature of this relationship is difficult to infer from this study alone, as the slow temporal resolution of fMRI cannot establish the directionality between the hippocampus and the visual-perceptual cortex. This is an exciting future avenue to explore.

We thank the reviewer for highlighting our contributions and suggesting that the relationship between visual imagery and autobiographical memory recall is an exciting future avenue.

Weaknesses:A weakness of the study is that some of the questions used are a bit vague, and no objective measure is used, which could have been more informative. For example, the spatial navigation question (reported as 'How difficult is it typically for you to orient you spatially?' - a question which is ungrammatical, but potentially reflects a typo in the manuscript) could have been more nuanced to tap into whether participants relied mostly on cognitive maps (likely supported by the hippocampus) or landmarks. It would also have been interesting to conduct a spatial navigation task, as participants do not necessarily have insight into their spatial navigation abilities (they could have been overconfident or underconfident in their abilities).Secondly, the question 'how difficult is it typically for you to use your imagination?' could also be more nuanced, as imagination is used in a variety of ways, and we only have reason to hypothesise that people with aphantasia might have difficulties in some cases (i.e. sensory imagination involving perceptual details). It is unlikely that people with aphantasia would have more difficulty than controls in using their imagination to imagine counterfactual situations and engage in counterfactual thought (de Brigard et al., 2013, https://doi.org/10.1016%2Fj.neuropsychologia.2013.01.015) due to its non-sensory nature, but the question used does not distinguish between these types of imagination. Again, this is a ripe area for future research. The general phrasing of 'how difficult is [x]' could also potentially bias participants towards more negative answers, something which ought to be controlled for in future research.

The main goal of our study was to examine autobiographical memory recall. Therefore, we used the gold standard Autobiographical Interview, or AI (Levine et al. 2002) and an fMRI paradigm to explore autobiographical memory recall as standardised, precisely, and objectively as possible.

In addition to these experimentally rigorous tasks, we employed some loosely formulated questions with the intention for people to reflect on how they perceive their own abilities to recall autobiographical memories, navigate spatially, and use their imagination. We agree with the reviewer that these questions are vague and did not have the experimental standard for an investigation into spatial cognition or imagination associated with aphantasia. Nonetheless, we believe that these questions provide important additional insights into what participants think about their own cognitive abilities. In order to set these questions into perspective, we argue in the discussion that spatial cognition and other cognitive functions should be investigated in more depth in individuals with aphantasia in the future.

As an additional note, all tasks were conducted in German. Thus, we were able to correct the wording of the debriefing question in our revision. We thank the reviewer for bringing this to our attention.

Strengths:A great strength of this study is that it introduces a fMRI paradigm in addition to the autobiographical interview, paralleling work done on episodic memory in cognitive science (e.g. Addis and Schacter, 2007, https://doi.org/10.1016%2Fj.neuropsychologia.2006.10.016), which has examined episodic and semantic memory in relation to imagination (future simulation) in non-aphantasic participants as well as clinical populations. Future work could build on this study, and for example use the recombination paradigm (Addis et al. 2009, 10.1016/j.neuropsychologia.2008.10.026), which would shed further light on the ability of people with aphantasia to both remember and imagine events. Future work could also build on the interesting findings regarding spatial navigation, which together with previous findings in aphantasia (e.g. Bainbridge et al., 2021, https://doi.org/10.1016/j.cortex.2020.11.014) strongly suggests that spatial abilities in people with aphantasia are unaffected. This can shed further light on the different neural pathways of spatial and object memory in general. In general, this study opens up a multitude of new avenues to explore and is likely to have a great impact on the field of aphantasia research.

We much appreciate the acknowledgment of our work into autobiographical memory employing both the autobiographical interview and fMRI. Furthermore, we hope that our work inspires future research in the way the reviewer outlines and in the way we describe in our manuscript.

**Reviewer #2 (Public Review):**
Summary:This study investigates to what extent neural processing of autobiographical memory retrieval is altered in people who are unable to generate mental images ('aphantasia'). Self-report as well as objective measures were used to establish that the aphantasia group indeed had lower imagery vividness than the control group. The aphantasia group also reported fewer sensory and emotional details of autobiographical memories. In terms of brain activity, compared to controls, aphantasics had a reduction in activity in the hippocampus and an increase in activity in the visual cortex during autobiographical memory retrieval. For controls, these two regions were also functionally connected during autobiographical memory retrieval, which did not seem to be the case for aphantasics. Finally, resting-state connectivity between the visual cortex and hippocampus was positively related to autobiographical vividness in the control group but negatively in the aphantasia group. The results are in line with the idea that aphantasia is caused by an increase in noise within the visual system combined with a decrease in top-down communication from the hippocampus.Recent years have seen a lot of interest in the influence of aphantasia on other cognitive functions and one of the most consistent findings is deficits in autobiographical memory. This is one of the first studies to investigate the neural correlates underlying this difference, thereby substantially increasing our understanding of aphantasia and the relationship between mental imagery and autobiographical memory.

We thank the reviewer for highlighting the importance of our findings.

Strengths:One of the major strengths of this study is the use of both self-report as well as objective measures to quantify imagery ability. Furthermore, the fMRI analyses are hypothesis-driven and reveal unambiguous results, with alterations in hippocampal and visual cortex processing seeming to underlie the deficits in autobiographical memory.

Once again, we thank the reviewer for highlighting the quality of our methods and our results.

Weaknesses:In terms of weaknesses, the control task, doing mathematical sums, also differs from the autobiographical memory task in aspects that are unrelated to imagery or memory, such as self-relevance and emotional salience, which makes it hard to conclude that the differences in activity are reflecting only the cognitive processes under investigation.

We agree with the reviewer that our control task differs from autobiographical memory in many different ways. In fact, for this first investigation of the neural correlates of autobiographical memory in aphantasia, this is precisely the reason why we chose this mental arithmetic (MA) task. We know from previous studies, that MA is, as much as possible, not dependent on hippocampal memory processes (Addis, et al. 2007, McCormick et al. 2015, 2017, Leelaarporn et al., 2024). The main goal of the current study was to establish whether there are any differences between individuals with aphantasia and controls. In the next investigation, we can now build on these findings to disentangle in more detail what this difference reflects.

Overall, I believe that this is a timely and important contribution to the field and will inspire novel avenues for further investigation.

This highly positive conclusion is much appreciated.

References

Addis, D. R., Wong, A. T., & Schacter, D. L. (2007). Remembering the past and imagining the future: Common and distinct neural substrates during event construction and elaboration. *Neuropsychologia*, *45*(7), 1363-1377.

Kriegeskorte, N., Simmons, W., Bellgowan, P. *et al.* Circular analysis in systems neuroscience: the dangers of double dipping. *Nat Neurosci*
**12**, 535–540 (2009). https://doi.org/10.1038/nn.2303

Leelaarporn, P., Dalton, M. A., Stirnberg, R., Stöcker, T., Spottke, A., Schneider, A., & McCormick, C. (2024). Hippocampal subfields and their neocortical interactions during autobiographical memory. *Imaging Neuroscience*.

Levine, B., Svoboda, E., Hay, J. F., Winocur, G., & Moscovitch, M. (2002). Aging and autobiographical memory: dissociating episodic from semantic retrieval. *Psychology and aging*, *17*(4), 677.

McCormick, C., St-Laurent, M., Ty, A., Valiante, T. A., & McAndrews, M. P. (2015). Functional and effective hippocampal–neocortical connectivity during construction and elaboration of autobiographical memory retrieval. *Cerebral cortex*, *25*(5), 1297-1305.

McCormick, C., Moscovitch, M., Valiante, T. A., Cohn, M., & McAndrews, M. P. (2018). Different neural routes to autobiographical memory recall in healthy people and individuals with left medial temporal lobe epilepsy. *Neuropsychologia*, *110*, 26-36.

**Recommendations for the authors:**

**Reviewer #1 (Recommendations For The Authors):**
This is a very interesting article that makes a substantial contribution to the field of the study of aphantasia as well as the neural mechanisms of autobiographical memory. I would strongly recommend this manuscript to be accepted (with these minor revisions), as it makes a substantial and well-evidenced contribution to the research, and it opens up many interesting avenues for researchers to explore. I was especially excited to see that the Autobiographical Interview had been paired with an fMRI paradigm, something which this field of research highly benefits from, as there are yet so few fMRI studies into aphantasia. I understand that it is the authors' decision whether to accept or reject any of the revisions I recommend here, but I would like to stress that I encourage accepting the recommended revisions, especially as there are some minor inaccuracies in the manuscript as it currently stands. Finally, I would like to stress that though I am based in the area of cognitive science, am not trained in fMRI imaging techniques, and therefore do not stand in a position where I can comment on the methodology pertaining to this part of the study - I encourage the Editors to seek a second reviewer's opinion on this.

Thank you for the positive evaluation of our manuscript as well as your comments. We have revised our manuscript according to your important suggestions as further explained below.

Line 33: "aphantasia prohibits people from experiencing visual imagery". This characterisation of aphantasia is too strong, especially as the authors use 32 as a cut-off point on the VVIQ, which represents weak and dim imagery. I would recommend using language like 'people with aphantasia have reduced visual imagery abilities', as this more accurately captures the group of people studied. Please revise throughout the manuscript. Please consult Blomkvist and Marks (2023) on this point who have discussed this problem in the aphantasia literature.

We agree that aphantasics may experience reduced visual imagery abilities. We have revised our wording throughout the manuscript.

Line 49: The authors conclude that their results 'indicate that visual mental imagery is essential for detail-rich, vivid AM', but this seems to be a bit too strong, for example since AM can be detail-rich with external (rather than internal) detail, and a person could potentially use mnemonic tricks such as keeping a detail-rich diary in order to boost their memory. That visual imagery is 'essential' implies that it is the only way to achieve detail-rich vivid AM, and this does not seem to be supported by the findings. I would recommend rephrasing it as 'visual mental imagery plays an important role in detail-rich, vivid AM' or 'visual mental imagery mediated detail-rich vivid AM'.

We altered the sentence in Line 49 using one of the recommended phrases:

‘Our results indicate that visual mental imagery plays an important role in detail-rich, vivid AM, and that this type of cognitive function is supported by the functional connection between the hippocampus and the visual-perceptual cortex.’

Line 69: Blomkvist and Marks (2023) have warned against calling aphantasia a 'condition' and this moreover seems to fit with the authors' previous research (Monzel, 2022). Please consider instead calling aphantasia an 'individual difference' in mental imagery abilities.

Thank you for the suggestion. We have revised our wording throughout the manuscript, avoiding the term ‘condition’.

Line 72: Add reference for emotional strength which has also been researched (Wicken et al. 2021, https://doi.org/10.1016/j.cortex.2020.11.014).

We have added the suggested reference in Line 75:

‘Indeed, a handful of previous studies report convergent evidence that aphantasics report less sensory AM details than controls (Bainbridge et al., 2021; Dawes et al., 2020, 2022; Milton et al., 2020; Zeman et al., 2020), which may also be less emotional (Monzel et al., 2023; Wicken et al., 2021).’

72-73: 'absence of voluntary imagery' - too strong as many people with aphantasia report having weak/dim mental imagery on the VVIQ.

We agree that aphantasics may experience reduced visual imagery. We have revised this notion throughout the manuscript.

74: Add reference to Bainbridge study which found a difference between recall of object vs spatial memory. This would be relevant here.

We have added the suggested reference in Line 76:

‘Spatial accuracy, on the other hand, was not found to be impaired (Bainbridge et al., 2021).’

Lines 94-97: The authors mention 'a prominent theory' but it is unclear which theory is referred to here. The article cited by Pearson (2019) does not suggest the possibility that aphantasia is due to altered connectivity between the hippocampus and visual-perceptual cortices. It suggests that aphantasia is due to impairment in the ventral stream, and in fact says that the hippocampus is unlikely to be affected due to spared spatial abilities in people with aphantasia. Specifically, Pearson claims: "Accordingly, memory areas of the brain that process spatial properties, including the hippocampus, may not be the underlying cause of aphantasia." (page 631). The authors further come back to this point in the discussion section (see comment below), saying that the hypothesis attributed to Pearson is supported by their study. I do not disagree with the point that the hypothesis is supported by the data, but it is unclear to me why the hypothesis is attributed to Pearson.

Thank you for pointing out this inaccuracy. We have edited the text to spell out our entire train of thought (see Lines 96-102):

‘A prominent theory posits that because of this hyperactivity, small signals elicited during the construction of mental imagery may not be detected (Pearson, 2019, Keogh et al., 2020). Pearson further speculates that since spatial abilities seem to be spared, the hippocampus may not be the underlying cause of aphantasia. In agreement, Bergmann and Ortiz-Tudela (2023) speculate that individuals with aphantasia might lack the ability to reinstate visually precise episodic elements from memory due to altered feedback from the visual cortex.’

Line 97: Blomkvist reference should be 2022 (when first published online).

The article ‘Aphantasia: In search of a theory’ by Blomkvist was first published on 1st July 2022. However, a correction was added on 13th March 2023. Therefore, we had cited the corrected version in this manuscript. However, we agree that the first publication date should be used and edited the reference accordingly.

Line 116: 'one aphantasic' could be seen as offensive. I would suggest 'one aphantasic participant'.

We have altered the paragraph according to your suggestion.

Line 138: In line with the recommendations put forward by Blomkvist and Marks (2023), I would suggest removing the word 'diagnosed', as this medicalises aphantasia in a way that is not consistent with its not being a kind of mental disorder (Monzel et al., 2022). I would say that aphantasia is instead operationalised as a score between 16-32. However, note that Blomkvist (2022) and Blomkvist and Marks (2023, https://doi.org/10.1016/j.cortex.2023.09.004) point out that there is also a lot of inconsistency in this score and how it is used in different studies. In your manuscript, I would recommend removing all wording that indicates that people with aphantasia have no experience of mental imagery, as you have operationalised for a score up to 32 which indicates vague and dim imagery. Describing vague and dim imagery as no imagery/absence of imagery is inconsistent (but common practice in the literature).

Thank you for your suggestion. We have revised the entire manuscript to eliminate any ambiguous meanings regarding the definition of aphantasia. Moreover, we replaced the word ‘diagnosed’ with ‘identified’ in Line 146.

Line 153: maybe 'correlated with imagery strength' rather than 'measures imagery strength'?

We have altered the sentence according to your suggestion in Line 160:

‘Previous studies have shown that the binocular rivalry task validly correlated with mental imagery strength.’

Line 162: "For participants who were younger than 34 years, the middle-age memory was replaced by another early adulthood memory". Is there precedence for this? Please add one sentence to explain/justify for the reader why a memory from this time period was chosen.

To maintain the homogeneous data set of acquiring five episodic autobiographical memories from five different periods of life per one individual, we asked the participants who were at the time of the interview, younger than 34 years old, to provide another early adulthood memory instead of middle age memory, as they had not reached the age range of middle age. According to Levine et al. (2002), younger adults (age < 34 years old) selected 2 events from the early adulthood period. Hence, all participants provided the last time period with memories from their previous year. We have added an additional explanation in this section in Line 170:

‘In order to acquire five AMs in every participant, the middle age memory was replaced by another early adulthood memory for participants who were younger than 34 years old (see Levine et al., 2002). Hence, all participants provided the last time period with memories from their previous year.’

Line 169: "During the general probe, the interviewer asked the participant encouragingly to promote any additional details." Consider a different word choice, 'promote' sounds odd.

We have altered the sentence according to your suggestion in Line 180:

‘During the general probe, the interviewer asked the participant encouragingly to provide any additional details.’

Line 196-198: the phrasing of these questions could have biased participants toward reporting it being more difficult. Did the authors control for this possibility in any way? The phrasing ‘How easy is it for you to [x]?’ might also be considered in a future study.

Thank you for pointing this out. These debriefing questions were thought of as open questions to get people to talk about their experiences. They were not meant as rigorous scientific experiments. Framing it in a positive way is a good idea for future research.

We have edited the manuscript on Line 394-396:

‘The debriefing questions were employed as a way for participants to reflect on their own cognitive abilities. Of note, these were not meant to represent or replace necessary future experiments.’

Line 197: This question is ungrammatical. Is this a typo, or was this how the question was actually posed? What language was the study conducted in?

All interviews within this study were conducted in German. Hence, the questions listed in this current manuscript were all translated from German into English. We have added this information in the Materials and Methods section in Line 169 as well as restructured the referred questions from Line 208-210:

‘All interviews were conducted in German.’

(1) Typically, how difficult is it for you to recall autobiographical memories?

(2) Typically, how difficult is it for you to orient yourself spatially?

(3) Typically, how difficult is it for you to use your imagination?’

Line 211: The authors write that participants were asked to "re-experience the chosen AM and elaborate as many details as possible in their mind's eye" was this the instruction used? I think stating the explicit instruction here would be relevant for the reader. If this is the word choice, it is also interesting as the autobiographical interview does not normally specify to re-experience details 'in one's mind's eye'.

The instructions gi‘en to ’he par’Icipa’ts were to choose an AM and re-experience/elaborate it in their mind with as many details as possible without explaining them out loud. We have clarified this in Lines 221-223.

‘For the rest of the trial duration, participants were asked to re-experience the chosen AM and try to recall as many details as possible without speaking out loud.’

Line 213: Were ‘vivid’ and ‘faint’ the only two options? Why was a 5-point scale (like the VVIQ scale) not used to better be able to compare?

During the scanning session, the participants were given a button box which contained two buttons with 'vivid' by pressing the index finger and 'faint' by pressing the middle finger. The 5-point scale was not used to avoid confusion with the buttons during the scanning session. We have clarified this in Line 224:

‘We chose a simple two-button response in order to keep the task as easy as possible.’

Line 347: Do the authors mean the same thing by 'imagery strength' and 'imagery vividness'? This would be good to clarify as it is not clear that these words mean the same thing.

Imagery strength is often used to describe the results of the Binocular Rivalry Task, whereas vividness of mental imagery is often used to describe the results of the VVIQ. Although both tasks are correlated, the VVIQ measures vividness, whereas the dimension of the Binocular Rivalry Task is not clearly defined. We added this information in a footnote on page 10.

Lines 353 - 356: When the authors first say that aphantasics described fewer memory details than controls, does this refer to external + internal details? Please clarify.Lines 353-360: The authors first say that aphantasics report "internal details (M = 43.59, SD = 17.91) were reported more often than external details (M = 20.64, SD = 8.94)" (line 355). But then they say: "a 2-way interaction was found between the type of memory details and group, F(1, 27) = 54.09, p < .001, ηp2 = .67, indicating that aphantasics reported significantly less internal memory details, t(27) = 5.07, p < .001, d = 1.83, but not significantly less external memory details, t(27) = 0.13, p = .898, compared to controls (see Figure 1b)" (line 358). This seems to first say that aphantasics didn't report fewer details than controls, but then that they did report fewer internal details than controls. Please clarify if this is correct.Line 383: Results from controls are not reported in this section.

We have first reported the main effects of the different factors; thus, aphantasics reported less details than controls (no matter of group and type of memory details), the internal details were reported more often than external details (no matter of group and memory period), and more details were reported for recent than remote memories (no matter of group and type of memory details). Subsequently, we report the simple effects for aphantasics and controls separately. To further clarify, we added the following segment in line 360:

‘Regarding the AI, we found significant main effects of memory period, F(1, 27) = 11.88, p = .002, ηp2 = .31, type of memory details, F(1, 27) = 189.03, p < .001, ηp2 = .88, and group, F(1, 27) = 9.98, p = .004, ηp2 = .27. When the other conditions were collapsed, aphantasics (M = 26.29, SD = 9.58) described less memory details than controls (M = 38.36, SD = 10.99). For aphantasics and controls combined, more details were reported for recent (M = 35.17, SD = 14.19) than remote memories (M = 29.06, SD = 11.12), and internal details (M = 43.59, SD = 17.91) were reported more often than external details (M = 20.64, SD = 8.94). More importantly, a 2-way interaction was found between type of memory details and group, F(1, 27) = 54.09, p < .001, ηp2 = .67, indicating that aphantasics reported significantly less internal memory details, t(27) = 5.07, p < .001, d = 1.83, but not significantly less external memory details, t(27) = 0.13, p = .898, compared to controls (see Figure 1b).’

Overall, the results were reported for aphantasics and controls separately in Lines 368-372.

Line 386: The question does not specify that it's asking about using imagination in daily life, even though this is what results report. I'm not sure that the question implies the use of imagination in daily life, so I would recommend removing this reference here.

We have removed the “in daily life” since this was not part of the original debriefing question.

Line 394: Could this slowness in response reflect uncertainty about the vividness?

Since the reason for this slowness is not known, we have refrained from adding this to the discussion. However, we added this as a short insertion in line 406:

‘Moreover, aphantasics responded slower (M = 1.34 s, SD = 0.38 s) than controls (M = 1.00 s, SD = 0.29 s) when they were asked whether their retrieved memories were vivid or faint, t(28) = 2.78, p = .009, possibly reflecting uncertainty in their response.’

Line 443: Graph E, significance not indicated on the graph.

After preprocessing, the fMRI data were statistically analyzed using the GLM contrast AM versus MA. The resulting images were then thresholded at p < 0.001, so that the illuminated voxels in Fig. 3 A, B, C, and D show only voxel in which we know already that there is a statistical difference between our conditions. Graph E illustrates only the descriptive means and variance of the significant differences in Fig. 3 C and D. This display is useful since the reader can more easily assess the difference between two conditions and two groups at a glance. For a general discussion on this topic, please also see circular analysis in fMRI (Kriegeskorte et al. 2009)

Line 521-522: The authors claim that Pearson (2019) forwards the hypothesis that heightened activity of visual-perceptual cortices hinders aphantasics from detecting small imagery-related signals. However, I find no statement of this hypothesis in Pearson (2019). It is unclear to me why this hypothesis is attributed to Pearson (2019). Please remove this reference or provide a correct citation for where the hypothesis is stated. Further, it is not clear from what is written how the results support this hypothesis as this is rather brief - please elaborate on this.

We attributed this hypothesis to Pearson (2019) according to his Fig. 4, which states: ‘A strong top-down signal and low noise (bottom left) gives the strongest mental image (square), whereas a high level of neural noise and a weak top-down imagery signal would produce the weakest imagery experience (top right).’

We have edited our manuscript to reflect Pearson better in Lines 543-550:

‘In a prominent review, Pearson synthesizes evidence about the neural mechanism of imagery strength (Pearson, 2019). Indeed, activity metrics in the visual cortex predict imagery strength (Cui et al., 2007; Dijkstra et al., 2017). Interestingly, lower resting activity and excitability result in stronger imagery, and reducing cortical activity in the visual cortex via transcranial direct current stimulation (tDCS) increases visual imagery strength (Keogh et al., 2020). Thus, one potential mechanism of aphantasia-related AM deficits is that the heightened activity of the visual-perceptual cortices observed in our and previous work hinders aphantasics to detect weaker imagery-related signals.’

Line 575: Consider citing Blomkvist (2022) who has argued that aphantasia is an episodic memory condition

We added the suggested reference in Line 601.

Line 585: Consider citing Bainbridge et al (2021) https://doi.org/10.1016/j.cortex.2020.11.014

We have added the suggested reference in Line 612.

Line 581: It might be relevant here to also discuss non-visual details, which have indeed been investigated in your present study. E.g. the lower emotional details, temporal details, place details, etc.

We have edited our discussion to reflect the non-visual details better in Line 605:

‘In fact, previous and the current study show that aphantasics and individuals with hippocampal damage report less internal details across several memory detail subcategories, such as emotional details and temporal details (Rosenbaum et al., 2008; St-Laurent et al., 2009; Steinvorth et al., 2005), and these deficits can be observed regardless of the recency of the memory (Miller et al., 2020). These similarities suggest that aphantasics are not merely missing the visual-perceptual details to specific AM, but they have a profound deficit associated with the retrieval of AM.’

Place details are discussed on page 37 onwards.

Line 605: I agree with this interesting suggestion for future research. It would also be relevant to reference Bainbridge (2021) here who tested spatial cognition in a drawing task and found that aphantasic participants correctly recalled spatial layouts of rooms but reported fewer objects than controls. It might also be worth pointing out that the present study does not actually test for accuracy in spatial cognition, so it could be the case that people with aphantasia feel confident that they can navigate well, but they might in fact not. Future studies relying on objective measures should test this possibility.

We have added the suggested reference in Line 625.

Lines 609-614: Is there any evidence that complex decision-making and complex empathy tasks depend on constructed scenes with visual-perceptual details? This hypothesis seems a bit far-fetched without any supporting evidence. In fact, it seems unlikely to be supported as we also know that people with aphantasia generally live normal lives, and often have careers that we can assume involve complex decision-making (see Zeman 2020 who report aphantasics who work as computer scientists, managers, etc). I would recommend that the authors provide evidence of the role of mental imagery in complex decision-making and complex empathy tasks, mediated by scene construction, to support this hypothesis as viable to test for future research. It is also unclear how this point connects to the argument made by Bergmann and Ortiz-Tudela (2023). In fact, Bergmann and Ortiz-Tudela seem to make the same argument as Pearson (2019) does - that aphantasia results from impairments in the ventral stream, but that the dorsal stream is unaffected. However, Blomkvist (2022) argues that this view is too simplistic to be able to account for the variety of deficits that we see in aphantasia. I would recommend either engaging more fully with this debate or cutting it, as it currently is too vague for a reader to follow.

We have decided to leave the discussion about scene construction and its connection to complex decision making and empathy out of the current manuscript. We have included the argument of Bergmann & Ortiz-Tudela (2023) in the Introduction (Line 101):

‘In agreement, Bergmann and Ortiz-Tudela (2023) speculate that individuals with aphantasia might lack the ability to reinstate visually precise episodic elements from memory due to altered feedback from the visual cortex.’

**Reviewer #2 (Recommendations For The Authors):**
In general, I really enjoyed reading this paper.

Thank you very much for the positive evaluation of our manuscript as well as your comments.

There were only a few things that I had some concerns about. For example, it was unclear to me whether the whole-brain analysis (Figures 3 and 4) was corrected for multiple comparisons or why only a small volume correction was applied for the functional connectivity analysis. If these results are borderline significant, this should be made more explicit in the manuscript. I don't think this is a major issue as the investigation of both the hippocampus and visual cortex was strongly hypothesis-driven, but it would still be good to be explicit about the strength of the findings.

For the whole-brain analysis, we applied a threshold of *p* < .001, voxel cluster of 10, but no other multiple comparisons correction applied. The peak in the right hippocampus did survive the whole-brain threshold but we decided to lower this threshold just for display purposes in Figure 3, so that the readers can easily see the cluster.

We have made the statistical thresholds more easily assessable for the reader on the following pages:

Figure 3 (Page 27): ‘Images are thresholded at p < .001, cluster size 10, uncorrected, except (D) which is thresholded at p < .01, cluster size 10, for display purposes only (i.e., the peak voxel and adjacent 10 voxels also survived p < .001, uncorrected).’

Figure 4 (Page 30): ‘Image is displayed at p < .05, small volume corrected, and a voxel cluster threshold of 10 adjacent voxels.’

I was wondering whether it would be possible to use DCM to investigate the directionality of the connectivity. Given that there are only two ROIs and two alternative hypotheses (top-down versus bottom-up) this seems like an ideal DCM problem.

We thank the reviewer for this suggestion and will consider testing the effective connectivity between both regions of interest in a future investigation.

Line 385: typo: 'great' should be 'greater'.

We have altered the typo from ‘great’ to ‘greater’ in Line 397.

Line 400: absence of evidence of an effect is not evidence of absence of an effect.

We agree with the reviewer that this was unclear. We changed the wording in Line 412:

‘In addition, aphantasics and controls did not differ significantly in their time searching for a memory in AM trials, t(19) = 1.03, p = .315.’

Typo line 623: 'overseas'.

We have altered the mistyped word from ‘overseas’ to ‘oversees’ in Line 647.